# FER-PCVT: Facial Expression Recognition with Patch-Convolutional Vision Transformer for Stroke Patients

**DOI:** 10.3390/brainsci12121626

**Published:** 2022-11-28

**Authors:** Yiming Fan, Hewei Wang, Xiaoyu Zhu, Xiangming Cao, Chuanjian Yi, Yao Chen, Jie Jia, Xiaofeng Lu

**Affiliations:** 1School of Communication and Information Engineering, Shanghai University, Shanghai 200444, China; 2Department of Rehabilitation, Huashan Hospital, Fudan University, Shanghai 200040, China; 3Department of Oncology, Jiangyin People’s Hospital Affiliated to Nantong University, Wuxi 214400, China; 4Department of Rehabilitation, The Affiliated Hospital of Qingdao University, Qingdao 266000, China; 5Department of Rehabilitation, Shanghai Third Rehabilitation Hospital, Shanghai 200436, China; 6Wenzhou Institute, Shanghai University, Wenzhou 325000, China

**Keywords:** facial expression recognition (FER), vision transformer (ViT), convolutional neural networks (CNNs), stroke, rehabilitation

## Abstract

Early rehabilitation with the right intensity contributes to the physical recovery of stroke survivors. In clinical practice, physicians determine whether the training intensity is suitable for rehabilitation based on patients’ narratives, training scores, and evaluation scales, which puts tremendous pressure on medical resources. In this study, a lightweight facial expression recognition algorithm is proposed to diagnose stroke patients’ training motivations automatically. First, the properties of convolution are introduced into the Vision Transformer’s structure, allowing the model to extract both local and global features of facial expressions. Second, the pyramid-shaped feature output mode in Convolutional Neural Networks is also introduced to reduce the model’s parameters and calculation costs significantly. Moreover, a classifier that can better classify facial expressions of stroke patients is designed to improve performance further. We verified the proposed algorithm on the Real-world Affective Faces Database (RAF-DB), the Face Expression Recognition Plus Dataset (FER+), and a private dataset for stroke patients. Experiments show that the backbone network of the proposed algorithm achieves better performance than Pyramid Vision Transformer (PvT) and Convolutional Vision Transformer (CvT) with fewer parameters and Floating-point Operations Per Second (FLOPs). In addition, the algorithm reaches an 89.44% accuracy on the RAF-DB dataset, which is higher than other recent studies. In particular, it obtains an accuracy of 99.81% on the private dataset, with only 4.10M parameters.

## 1. Introduction

The incidence, mortality, and disability of stroke in China have been higher than those in developed countries such as the United Kingdom, the United States, and Japan in the past 15 years [1]. Most stroke survivors cannot normally live because of suffering from sequelae such as hemiplegia, limb numbness, swallowing disorders, and depression. Brain neurobiology suggests that early training, at the right intensity, will aid recovery [2]. However, physicians need to be aware of patients’ feelings in real-time during early rehabilitation to determine whether the training matches their physical recovery, then tailor the most rehabilitation-friendly training for each patient. This existing manual monitoring mode in clinical practice, which causes an enormous burden on medical resources, urgently needs to be improved and optimized.

Deep learning, as one of the powerful medical assistance technologies, has been widely applied in the medical field [3]. These applications include but are not limited to automatic diagnosis of breast cancer based on whole slide imaging [4], accurate measurement of morphological changes in intervertebral discs based on axial spine Magnetic Resonance Image (MRI) [5], detection of fundus lesions based on fundus imaging [6], and segmentation of brain tumors based on T1-weighted MRI [7]. These studies show that deep learning dramatically reduces the heavy and urgent workload of physicians and improves the efficiency of medical care. In particular, deep learning and machine learning also play an important role in stroke prediction, prognostics, and management. Iqram and Se proposed a real-time health monitoring system for stroke prognostics [8] and a cardiac monitoring system for stroke management [9]. These studies provide technical support for early stroke prognostics and have medical practice implications for predicting acute stroke. In addition, emotion classification techniques for stroke patients based on electroencephalography (EEG) [10,11,12] and facial electromyography (EMG) [13] also provide physicians with meaningful assessment information to replace traditional clinical methods based on observation or scoring.

However, these physiological signal-based approaches to stroke prognostics inevitably require contact with the patient’s skin to capture the information. For stroke rehabilitation, wearable devices are likely to interfere with patients’ training. In contrast, the facial expression recognition (FER) technique based on computer vision can acquire the state of patients in training without contact, which is more suitable for stroke rehabilitation. At present, although the application of the FER technique in the field of stroke rehabilitation is less than that of other medical fields, such as Down syndrome prediction [14], depression diagnosis [15], and autism spectrum disorder identification [16], it is significantly improving stroke management and quality of stroke care [17,18].

Few studies have published facial expression datasets and FER algorithms for stroke patients because of their privacy and sensitive nature. However, FER for healthy people is one of the mainstream tasks in computer vision. The relevant datasets contain a large number of samples, such as the Facial Expression Recognition 2013 dataset (FER2013, 35,886 images) [19], the Static Facial Expression in the Wild (SFEW, 1766 images) [20], the Real-world Affective Faces Database (RAF-DB, 29,672 images) [21], and AffectNet (400,000 images) [22]. In terms of algorithms, Convolutional Neural Networks (CNNs), such as Visual Geometry Group (VGG) [23], Google Inception Network (GoogleNet) [24], and Residual Network (ResNet) [25], are the most commonly used structures in this field because of their excellent robustness to changes in face position and image scale. With Attention [26] proposed, many studies used it to replace part of CNNs or combine them to improve performance while the model’s overall structure remained unchanged [27,28,29]. In 2020, Vision Transformer (ViT) did not use convolution and outperformed state-of-the-art CNNs on mainstream small- and med-sized image classification datasets [30]. Moreover, in recent years, improved algorithms based on Transformer have continuously surpassed the previous algorithms in performance [31,32]. Fayyaz et al. [33] showed that ViT is more robust than CNNs in handling occluded images, feature transformation, and token reorganization. Nevertheless, the proposers of ViT also illustrated that ViT outperforms ResNet only when trained on enormous datasets (14–300 million images). To address this problem, some studies have combined CNNs and Transformers to model both local and global dependencies for image classification [34,35]. However, these algorithms are designed to achieve higher accuracy, inevitably requiring huge parameters, computational cost, and Giga Floating-point Operations Per Second (GFLOPs), which make them hard to embed into rehabilitation medical equipment.

We aim to design a lightweight FER algorithm for stroke rehabilitation in clinical practice, so as to assist physicians in determining whether the training intensity of stroke patients matches their physical rehabilitation and whether patients are active or focused during training. The key contributions of this paper can be summarized as follows:We propose a lightweight FER algorithm named Facial Expression Recognition with Patch-Convolutional Vision Transformer. It requires less memory and computation for model training/inference while ensuring high accuracy.The proposed algorithm effectively combines the local perception ability of CNN and the advantages of ViT in extracting global features, which makes the algorithm achieve the highest accuracy on the RAF-DB dataset.We treat emotion features as the weighted sum of neutral and V-A-like emotion features at different scales and design a unique classifier, which has been verified that more detailed facial emotion information of stroke patients has been extracted for classification.

## 2. Materials and Methods

### 2.1. Data Sources and Data Preprocessing

There are three datasets used in this study: (1) two public datasets for healthy people, RAF-DB [21] and FER+ [36]; (2) a private dataset for stroke patients. Table 1 describes the sample properties of three datasets in detail.

RAF-DB dataset

The Real-world Affective Faces Database (RAF-DB) contains a single-label subset with 15,339 images, which can be divided into seven basic emotional classes: happy, sad, surprised, angry, fearful, disgusted, and neutral. These samples are of significant variability in subjects’ age, ethnicity, head poses, lighting conditions, occlusions (e.g., glasses, facial hair, or self-occlusion), and post-processing operations (e.g., various filters and effects) [21]. These diverse differences make the trained models have better generalization.

FER+ dataset

The Face Expression Recognition Plus dataset (FER+) contains 35,887 images of size 48 × 48 that can be divided into 10-class emotions. Only 21,161 images/8 emotions are used in this experiment: happy, sad, surprised, angry, fearful, disgusted, neutral, and contempt.

Private dataset

The inclusion criteria were as follows: (1) patients aged 18–85 years old; (2) diagnosed with stroke confirmed by computed tomography (CT) and/or magnetic resonance imaging (MRI); (3) ≥2 weeks post-stroke; (4) upper limb of the healthy or affected side can use the upper limb rehabilitation robot for training; (5) patients signed the informed consent. The exclusion criteria were: (1) patients with unstable cerebrovascular disease; (2) patients with sensory aphasia or motor aphasia, and those who were unable to cooperate with assessment and testing; (3) Montreal Cognitive Assessment (MoCA) score ≤ 25; (4) patients with severe organ dysfunction or with malignant tumors; (5) House–Brackmann (H-B) grade ≥ III.

There were 42 participants in the experiment, of which 37 patients with stroke (25 men and 12 women, 31–87 years old) were confirmed cases from the Shanghai Third rehabilitation hospital and 5 healthy controls (4 physicians and 1 student). All subjects signed an informed consent form before the experiment.

In this study, four basic emotions (happy, sad, surprised, and angry) were used as biomarkers to assess the patient’s concentration, and four special emotions (painful, strained, tired, and neutral) were used as biomarkers to determine whether the current training intensity is suitable for the patient. There were two schemes for collecting emotional videos. First, we guided patients to express these four basic emotions through videos and pictures. Second, these four special emotions were collected while patients were training with the upper limb rehabilitation robot. In addition, we asked patients to repeatedly lift the upper extremity and gradually increase the range of motion to capture these desired emotions. In this experiment, each patient participated in collections of two emotions at least, which ensured that each subject’s sample had positive and negative labels.

After collecting the emotional videos, data preprocessing is an indispensable step, mainly sampling images, correcting faces, and labeling samples. The DB Face [37], a face detection algorithm, was used to predict the anchor boxes of faces and corresponding confidence scores in emotional videos automatically. Then, we removed face images with low confidence and incomplete from numerous video slices containing facial expressions. These preserved facial images were adjusted by rotating so that the line connecting the eyes’ feature points detected by the DB Face algorithm was in the horizontal direction, with the midpoint of the line as the center of rotation. The line’s rotation angle θ is calculated by Equation (1). The transformation matrix M of all pixels in the original image is defined as Equation (2). The coordinates of all original pixels can be transformed into the corrected coordinates using Equation (3).
(1)$$\theta =\tan^{-1}\frac{y_{r}-y_{l}}{x_{r}-x_{l}}$$
(2)$$A=\left[\begin{array}{cc} \cos\theta & \sin\theta \\ -\sin\theta & \cos\theta \end{array}\right],\;B=\left[\begin{array}{c} (1-\cos\theta )\cdot x_{c}-\sin\theta \cdot y_{c}\\ \sin\theta \cdot x_{c}+(1-\cos\theta )\cdot y_{c} \end{array}\right],\;M=\left[\begin{array}{cc} A & B \end{array}\right]$$
(3)$$\left[\begin{array}{c} x^{\prime }\\ y^{\prime } \end{array}\right]=A\left[\begin{array}{c} x\\ y \end{array}\right]+B=M\left[\begin{array}{c} x\\ y\\ 1 \end{array}\right]$$
where $\left(x_{l},y_{l}\right)$, $\left(x_{r},y_{r}\right)$, and $\left(x_{c},y_{c}\right)$ are the feature coordinates of the left eye, the right eye, and the midpoint of the line connecting eyes in the original image, respectively. $\left(x^{\prime },y^{\prime }\right)$ is the corrected coordinate.

We labeled the face-aligned images using the Facial Action Coding System (FACS) [38]. First, the emotional label of each sample was initially determined based on the content of the corresponding emotional video of the sample. Then, these images were annotated again according to FACS definitions of eight expressions. Table 2 shows FACS definitions of eight expressions in this experiment. In addition to the five expressions of happy, sad, angry, surprised, and neutral, the other expressions required for this experiment must be clearly defined by FACS. Referring to the PSPI [39], FACS features of painful expressions include lowered brow (AU4), raised cheeks (AU6), tightened lid (AU7), wrinkled nose (AU9), raised upper lip (AU10), and closed eyes (AU43). By comparing the facial features corresponding to each AU, we defined that FACS features of strained expressions are lowered brow (AU4), raised cheeks (AU6), tightened lips (AU23), pressed lips (AU24), and sucked lips (AU28); FACS features of tired expressions are closed eyes (AU43) and downed head (AU54), as shown in Figure 1.

After labeling and collation, the private dataset contains 1302 samples/8 categories, with no sample crossover and duplicates. Some samples of the private dataset are shown in Figure 2.

### 2.2. Model Building

In order to occupy fewer computing resources to identify eight facial expressions of stroke patients accurately, we propose a lightweight FER model shown in Figure 3, named the Facial Expression Recognition with Patch-Convolutional Vision Transformer (FER-PCVT). The FER-PCVT designed with ViT as the baseline mainly consists of three modules: the Convolutional Patch Embedding (CPE), the Pyramid Transformer (PTF), and the Valence-Arousal-Like Classifier (V-ALC). The first two modules combine to form the backbone network, Patch-Convolutional Vision Transformer (PCVT). The V-ALC is an expression classifier designed based on the Valence-Arousal (V-A) emotion theory [40].

#### 2.2.1. Convolutional Patch Embedding

Compared with the direct processing of pixel information of images using the transformer encoder of ViT, the accuracy will be further improved by using CNNs to extract the feature information from images and then processing them with the transformer encoder [35,41]. Based on this, the convolutional patch embedding module is implemented as a pixel-to-sequence mapping module to extract the feature sequences as the input of the Conv-TF Encoder of the pyramid transformer module. Specifically, the feature information extracted from the image by the convolutional layer and pooling layer is reduced to the patch size by the Block Scaling module. The Block Scaling module, consisting of two convolutional layers (size 2 × 2, stride 2, and size 1 × 1, stride 1), is applied to adjust the dimensions of feature maps entered into the Conv-TF Encoder by varying the number of repetitions. That is, the length and width of the sequence will be shortened to $1/2^{r}$ of the original size after repeating *r* times. This method of introducing convolutions into ViT achieves feature mapping from pixel to sequence while preserving the position information between patches. The detailed structure is shown in Figure 4a.

#### 2.2.2. Pyramid Transformer

ViT requires the input and output sequences in the transformer encoder to have the same dimensions. However, the length of sequences output by CNNs is reduced as the network deepens. This pyramidal output mode in the CNNs, significantly reducing the computational cost, has been shown to be beneficial in extracting feature information at different scales [42]. Thus, the PTF designed by introducing this output mode aims to reduce the storage, parameters, and GFLOPs required for computation. Details of the PTF are shown in Figure 4b. We use convolutional mapping instead of the linear mapping in the transformer encoder of ViT to extract the three feature matrices: Q, K, and V. Then, they are fed into the Multi-Head Self-Attention to be given different weights. In the Feed-Forward module, a bottleneck structure is formed by two convolutional layers with output channels $d_{i}/2$ and $d_{i}$, respectively, which compresses the channel dimension in the model. The activation function GeLU between the two convolutional layers is used to make the model fit data faster, and its expression is Equation (4).
(4)$$GeLU\left(x\right)=\frac{1}{2}x\left(1+erf\left(\frac{x}{\sqrt{2}}\right)\right)$$
where erf(·) is the Gauss Error Function. In the Block Combined Pooling module, $d_{0}$ convolution kernels (size 3 × 3) expand the channel dimension of the input feature map, followed by downsampling with a max pooling window (size 3 × 3, stride 2). The module allows feature maps to be resized from $d_{i}\times h_{i}\times w_{i}$ to $d_{0}\times h_{i}/2\times w_{i}/2$, gradually reducing the feature output, like a pyramid.

In addition, the layer normalization constrains the outputs of the Conv-TF Encoder module and the Feed-Forward module to avoid the vanishing gradient. The inputs and outputs of the above two modules are connected by residual connections to prevent the loss of feature information extracted by the model. At the same time, the batch normalization regularizes the output of the Block Combined Pooling module.

#### 2.2.3. Valence-Arousal-Like Classifier

FACS defines the neutral expression as no AU, meaning that no facial muscle movement can be used as a biomarker. It makes neutral expressions more challenging to identify than other expressions. Especially neutral expressions of some stroke patients are different from those of ordinary people when all facial muscles are completely relaxed. The V-A emotion theory [40] suggests that each emotion is a mixture of arousal and valence in different proportions. Referring to the theory, we design the V-ALC as an expression classifier, considering emotion as a weighted sum of neutral and V-A-like features. Details of the V-ALC are shown in Figure 5.

We adopt the pixel shuffle method to reshape low-resolution feature maps into high-resolution ones. That is, the length and width of the input feature map are up-sampled by 12 times, and the result is condensed using a convolution kernel of size 12 × 12. These compressed sequences are grouped into the Channel Mean and the Batch Sharing to obtain the V-A-like and neutral features with one dimension, respectively. The result of multiplying the neutral feature with the adaptive weight $w_{AD}$ is added to the V-A-like feature to output a complete feature map of emotion. Among them, $w_{AD}$ is a parameter learned by the model from many training samples. The Channel Mean means averaging the values of different channels in the same batch, thereby reducing the channel dimension. The Batch Sharing refers to averaging the values in different batches on the basis of the Channel Mean, which aims to extract the most appropriate characteristics of neutral emotions from batches. Their expressions are Equations (5) and (6).
(5)$$Channel\;Meal\;\left(x_{b}\right)=\frac{1}{c}\;\overset{c}{\underset{i=1}{\sum }}x_{bi}$$
(6)$$Batch\;Sharing\;\left(x\right)=\frac{1}{bc}\overset{b}{\underset{j=1}{\sum }}\overset{c}{\underset{i=1}{\sum }}x_{ij}$$
where x is the input feature tensor, $x_{b}$ is the feature sequence of different batches in the input tensor, i is the $i^{th}$ channel, j is the $j^{th}$ batch, c is the total number of channels, and b is the total number of batches.

After outputting a complete feature map of emotion, considering that emotion may be a composite state, we normalize these sequences using the Sigmoid function to avoid mutually exclusive results using the Softmax function. Finally, the prediction confidence of each category is output, where the expression with the highest confidence is the final result of the model’s prediction.

## 3. Results

### 3.1. Setup

Table 3 shows the training settings in this experiment, including the selected optimizer, the loss function, and some specific hyperparameters. Table 4 shows the detailed structural parameters for each module combined in this experiment.

### 3.2. Performance Evaluation of PCVT Based on Public Datasets

We evaluate the learning capabilities of CvT [35], PvT [42], ResNet18 [25], ResNet18*, and PCVT on the RAF-DB dataset, focusing on accuracy and resource consumption. Among them, both CvT and PvT are hybrid variant networks formed by introducing convolution into ViT, which are of the same type as this study. ResNet 18 is the most commonly used convolutional neural network for image classification, and ResNet18* is a pre-trained model of ResNet18. CvT, PvT, ResNet18, and PCVT are retrained from scratch using the same computer to obtain experimental results that are not affected by the device conditions. For ResNet18*, we further trained it using this emotion dataset on top of the parameter weights.

As shown in Figure 6, the iterative curves of these five networks trained and validated on the RAF-DB dataset show that the PCVT proposed in this study performs better on the validation data than other models except for ResNet18*. It means that PCVT has better generalization than PvT, CvT, and ResNet18. Admittedly, as a pre-trained model, ResNet18* predictably shows the best classification ability from the beginning of the iteration. Compare the parameters, GFLOPs, and accuracy of the above five networks on the RAF-DB dataset, as shown in Table 5. The accuracy of PCVT is 84.22%, second only to that of ResNet18* (86.28%). Meanwhile, PCVT has the fewest parameters and GFLOPs.

### 3.3. Performance Evaluation of FER-PCVT Based on Public Datasets

#### 3.3.1. Comparison with State-of-the-Art Methods

The proposed FER-PCVT is compared with the state-of-the-art methods on RAF-DB and FER+ datasets. As shown in Table 6, two FER-PCVT models without pretrained weights trained from scratch on two public datasets achieved 89.44% and 88.21% accuracy, respectively. The FER-PCVT learned on the RAF-DB achieves the highest accuracy, while the FER-PCVT learned on FER+ performs lower than other models.

#### 3.3.2. Analysis Based on Confusion Matrix

The detailed performance of FER-PCVT for each class on the RAF-DB and FER+ datasets is analyzed based on the confusion matrix. As shown in Figure 7, FER-PCVT is sensitive to whether the dataset is balanced. There is no significant deviation in the predicted results on the RAF-DB dataset. However, the model shows significant bias on the FER+ dataset. As shown in Figure 7b, the model’s predictions have extreme errors in the “disgust” and “contempt” classes with small samples; conversely, the model has highly accurate for the “happy” and “neutral” classes. Moreover, V-ALV determines the expression baseline based on the features of neutral expressions in the batch, so the imbalance of dataset affected the expression baseline generation. In addition, the Precision, Specificity, Sensitivity, F1-Score, and G-mean of FER-PCVT are also analyzed based on the confusion matrix, as shown in Table 7. We set the Precision and Recall to the same weight to obtain the F1-Score of FER-PCVT for each emotional category. On the RAF-DB dataset, the F1-Score values of FER-PCVT for surprised, fear, disgust, happy, sad, angry, and neutral are 84.2%, 73.3%, 67.9%, 94.5%, 86.7%, 80.4%, and 92.4%, respectively. However, on the FER+ dataset, FER-PCVT only performs well for categories with many samples, such as surprised (86.4%), happy (89.4%), sad (69.1%), and neutral (73.6%). G-mean reflects the contribution of each category to the model’s accuracy. Although the model’s accuracy reaches 88.21% on the FER+ dataset, the G-mean values of both disgust and contempt are 0%, which means that the accuracy depends on surprised (89.8%), fear (72.3%), happy (94%), sad (75.8%), angry (78.8%), and neutral (86.5%). In contrast, the G-mean values of all categories are higher than 90% in the RAF-DB dataset, and the order from high to low is neutral (96.8%), happy (95.6%), sad (92.1%), surprised (89.4%), anger (85.2%), fear (81.1%), and disgust (81.1%).

The above parameters for evaluating performance are calculated using the standard formulas shown in Equations (7)–(11):(7)$$Precision=\frac{TP}{TP+FP}$$
(8)$$Specificity=\frac{TN}{TN+FP}$$
(9)$$Sensitivity=Recall=\frac{TP}{TP+FN}$$
(10)$$F1-Score=\frac{2\times Precision\times Recall}{Precision+Recall}$$
(11)$$G_{mean}=\sqrt{Recall\times Specificity}$$
where *TP*, *TN*, *FP*, and *FN* mean the true positive, the true negative, the false positive, and the false negative, respectively.

#### 3.3.3. Visualization of Clustering Ability

The clustering ability of FER-PCVT on the RAF-DB dataset is visualized by the t-SNE plot based on the inputs and outputs of the last linear layer of V-ALC. As shown in Figure 8, the boundaries between the various categories are clear and intuitive, which means that FER-PCVT can distinguish and cluster the seven emotions well.

### 3.4. Performance Evaluation of FER-PCVT Based on the Private Dataset

#### 3.4.1. Accuracy Comparison and Impact of Pretrained Weights

We compare FER-PCVT with ResNet18 and the structure combining PCVT with the Multi-layer Perceptron (MLP) on the private dataset, focusing on the accuracy and parameters of these models with and without pretrained weights. Figure 9 shows the training and validation accuracy curves of ResNet18, PCVT+MLP, and FER-PCVT on the private dataset. As shown in Table 8, the structure formed by PCVT combined with MLP exhibits the worst precision on the private dataset, although it has the lowest number of parameters. FER-PCVT has similar accuracy to ResNet18 on the private dataset with or without pre-trained weights. However, the algorithm proposed in this experiment has only 4.10M parameters, about one-third of the parameters of ResNet18.

#### 3.4.2. Visualization of Clustering Ability

To visualize the model’s ability to classify the eight facial expressions of stroke patients, we plot the t-SNE of FER-PCVT on the private dataset. As shown in Figure 10, the model can cluster the four basic expressions and four special expressions of stroke patients well. Especially in special categories, the distribution of neutral expressions with other expressions is similar to that of the V-A emotion theory.

### 3.5. Visual Analysis

We perform a global visual analysis of models to find the regions that models focus on for classification. The Grad-CAM [52] is used to visualize ResNet18*. For ViT and FER-PCVT, visualization is achieved by stacking the attention weights of each layer in order. ResNet18*, ViT, and FER-PCVT have different focus points when identifying the facial emotions of stroke patients, as shown in Figure 11. The part covered in red is the region of the model’s most concern when classifying and recognizing expressions. ResNet18* focuses on localized facial regions, while ViT extracts information globally. Although the red regions in the visualization images of ViT appear more on the periphery of the image, ViT also pays attention to the details of the facial features. However, FER-PCVT can focus more on muscle changes due to different expressions while extracting global information. For example, for strained expression, a common emotion when muscles are tense during training, FER-PCVT notices more changes in areas such as eyebrows, eyes, and lips than in other models. Moreover, the facial features of neutral expressions extracted by FER-PCVT are more specific than those of other models. In addition, FER-PCVT also showed a better ability to extract emotional features for these four basic expressions.

## 4. Discussion

Experienced physicians can determine stroke patients’ intervention strategies by observing their emotional changes [13]. Similarly, stroke rehabilitation systems based on deep learning/machine learning can also sense the patients’ emotions and provide training suggestions according to emotional changes. Currently, more researchers use the patients’ physiological signals as the information source of perceived emotion [10,11,12,13]. Few studies have designed FER algorithms for stroke rehabilitation. To assist physicians in analyzing the degree of physical recovery and adjusting the training intensity of stroke patients, we use eight common emotions of patients during rehabilitation as biometrics and design a lightweight FER algorithm. By detecting the positive emotions of stroke patients during rehabilitation, such as happy, surprised, and strained, patients’ training motivation and interest will be provided to physicians. When painful emotions are detected, it means that the training intensity exceeds the patient’s muscle tolerance, and the intensity should be adjusted in time to avoid secondary injuries. In addition, if negative emotions are detected frequently, such as sad, tired, and angry expressions, physicians must pay attention to patients’ mental health.

The FER algorithm proposed in this study is an automated assessment technology for stroke rehabilitation, which acquires the training status of patients in a non-contact way. ViT is the basic framework for algorithm design since the global modeling of images using ViT is critical to the emotional classification task, as shown in Figure 11. However, CNNs structures are better at extracting local and detailed information in expression images than ViT. Therefore, introducing the characteristics of CNNs into the ViT structure can improve performance and robustness while maintaining high accuracy and memory efficiency. ViT converts the pixel information (2D) in the patch into the feature sequence (1D) required by the encoder through linear projection and Patch Embedding. The position relationships between patches need to be learned through the Position Embedding module. However, the sequence extracted by convolution contains position information, which is the inductive bias property of convolution. Thus, the CPE module containing convolutional layers and pooling layers is designed to replace the linear projection, the Patch Embedding, and the Position Embedding in ViT. There are some studies that have also introduced convolution into ViT. For example, VTFF [34] extracts the information from the original and local binary pattern images using two ResNet18. Then, it flattens and linearizes feature information to obtain patches with features instead of patches with image blocks in ViT. This network achieves an accuracy of 88.14% on RAF-DB while containing a large number of parameters (51.8M). However, the algorithm proposed in this study performs better on RAF-DB with 1.3% higher accuracy than VTFF. CvT [35] divides transformers into multiple stages, constituting the transformers’ hierarchy. A convolutional token embedding module is added at the beginning of each stage, which is implemented as a convolutional projection to replace the linear projection before each self-attention in ViT. The algorithm proposed in this paper mainly realizes the convolutional mapping between pixels to sequences by combing convolution and pooling instead of linear projection in ViT. At the same time, the location information between patches is preserved. In contrast, we incorporate convolutional features in ViT more concisely. According to the experimental data in Table 5 and Figure 6, PCVT proposed in this study has higher accuracy and lower parameters than CvT on the RAF-DB dataset.

In addition, high accuracy and low parameters are necessary for a model to run well in rehabilitation equipment with less computing power than professional computers. Therefore, we designed the PTF module that introduced a pyramidal feature output mode to reduce parameters and GFLOPs, inspired by PvT [42]. PvT is proposed as a backbone model to serve downstream tasks in various forms, such as image classification, object detection, and semantic segmentation. Similar to this study, both PvT and FER-PCVT have reduced the sequence length of the transformer output as the network deepens, significantly decreasing computational overhead. Regarding implementation details, PvT splits the image/feature map into many patches (size of $P_{i}\times P_{i}$, where i is the $i^{th}$ stage), and then feeds each patch into the linear projection to obtain many feature sequences whose dimensions are $P_{i}$ times shorter than the input. However, we mainly down-sample the feature map by combining convolutional and pooling to get a feature map that the size is reduced by half each time. Validated by the experiments shown in Figure 6 and Table 5, the proposed algorithm has higher accuracy and requires about 3.79M lower parameters than PvT in model training/inference.

Furthermore, considering that some stroke patients have different facial expressions due to impaired facial muscles, we designed a classifier that is more suitable for the emotion classification of stroke patients to improve the accuracy further. We designed the V-ALC classifier based on the V-A emotion theory, treating emotion as the weighted sum of V-A-like and neutral features. The addition of V-ALC improves the model’s accuracy from 84.22% to 89.44%, as shown in Table 5 and Table 6. According to Table 8, the structure obtained by PCVT splicing V-ALC performs better than that obtained by PCVT splicing MLP in classifying the emotions of stroke patients.

We also visually analyze models to find the attention regions of ViT, ResNet18*, and FER-PCVT in classifying emotions and verify that FER-PCVT combines the advantages of the other two structures well. As shown in Figure 11, ResNet18, a typical CNNs structure, focuses on the facial regions that best represent emotions, similar to the areas humans notice when recognizing the emotions of stroke patients. For example, the tightened and open lips when angry, the wrinkled eyebrows when sad, the raised cheeks when strained, and the relaxed eyes and mouth when tired. Unlike ResNet18, ViT extracts global features while also paying attention to some facial regions located inside the image, especially for surprised and painful expressions. FER-PCVT extracts information globally like ViT but perceives more detailed facial regions than ViT, which means more details about emotions can be captured by FER-PCVT.

However, the algorithm proposed in this study recommends using a dataset with better balance for training, since the designed classifier sums neutral emotion features with weights with other emotion features for classification. Unbalanced sample sizes will affect the model’s ability to extract an unbiased emotion baseline. The RAF-DB dataset is more balanced than the FER+ dataset, so the proposed method achieves the highest accuracy on the RAF-DB dataset, as shown in Table 6 and Figure 7. However, its performance on the FER+ dataset is weaker than other FER algorithms, such as RAN [29], VTFF [34], SCN [47], and FER-VT [48].

To summarize, the proposed method has several advantages: (1) It achieves higher recognition accuracy than other existing FER algorithms on the RAF-DB dataset. (2) The network structure successfully combines the local perception of CNNs and the global extraction capability of ViT, which effectively improves the ability of the model to extract feature sequences used to classify patients’ emotions. (3) It has fewer parameters and GFLOPs than other algorithms, making it easier to embed in medical rehabilitation equipment with poorer computing performance than professional computers. Although the proposed method has shown lower consumption and better effectiveness on both public datasets and the private dataset, there are still some problems to be improved: (1) The algorithm performs better on the balanced dataset. Therefore, it is necessary to balance the sample size of each category in order to obtain unbiased prediction results. (2) The sample size of the private dataset used in this study is insufficient compared to public datasets, especially for painful and tired expressions. We hope to collect more clinical data to improve the model’s generalization. (3) This study only conducted a qualitative analysis of emotions and did not further classify each emotion. For example, painful emotions are divided into severe, moderate, and slight pain in detail. It is hoped that future research can bring more specific and quantitative rehabilitation recommendations for the early training of stroke patients.

## 5. Conclusions

This study proposes a lightweight FER algorithm, FER-PCVT, which is more conducive to embedding in medical rehabilitation equipment to determine whether the current training intensity received by a stroke patient is most suitable for his physical recovery. To verify the performance of FER-PCVT, we collect and annotate a private dataset of stroke patients containing 1302 samples, which can be divided into 8 classes: painful, strained, tired, neutral, happy, sad, angry, and surprised. This algorithm is compared with other FER algorithms on two public datasets (FER+ and RAF-DB) and a private dataset. The experimental results show that: (1) PCVT, the backbone network of FER-PCVT, achieves an accuracy of 84.22%, parameters of 2.46M, and GFLOPs of 0.12 on the RAF-DB dataset, which is better than CvT, PvT, and ResNet18. (2) FER-PCVT achieves 88.21% and 89.44% on the FER+ and RAF-DB datasets, respectively. Its performance exceeds that of other existing expression recognition algorithms on the RAF-DB dataset. (3) FER-PCVT achieves an accuracy of 99.81% on the private dataset, with only 4.10M parameters. (4) FER-PCVT effectively combines the local perceptual ability and the feature output mode of the CNNs and the global extraction capability of ViT, which significantly reduces the parameters and ensures recognition accuracy. This method has excellent performance on public and private datasets, providing an intuitive and efficient automated assessment technique for stroke patients to receive more suitable early training.

## Figures and Tables

**Figure 1 brainsci-12-01626-f001:**
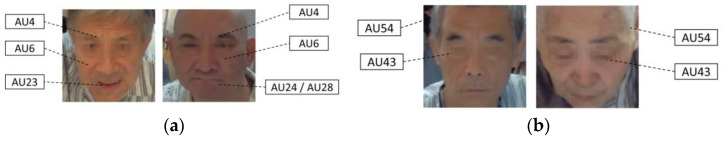
FACS features of strained and tired expressions: (**a**) the strained expressions spontaneously appeared by patients when their limb muscles were tense during training; (**b**) the tired expressions occurred when the patients were resting or undergoing prolonged passive training.

**Figure 2 brainsci-12-01626-f002:**
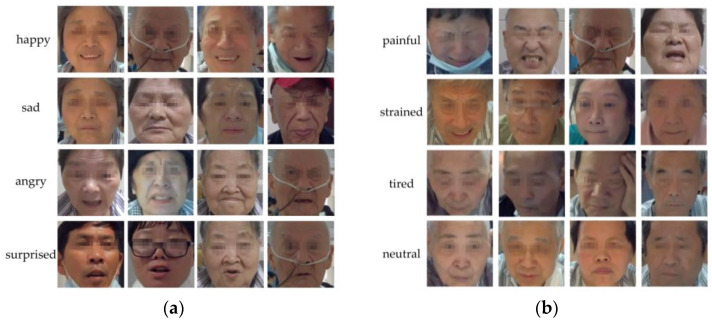
Partial samples of the private dataset: (**a**) these four basic expressions are used to assess training attention and positivity of stroke patients; (**b**) these four special expressions are used to determine whether the training intensity is proper for the patient.

**Figure 3 brainsci-12-01626-f003:**
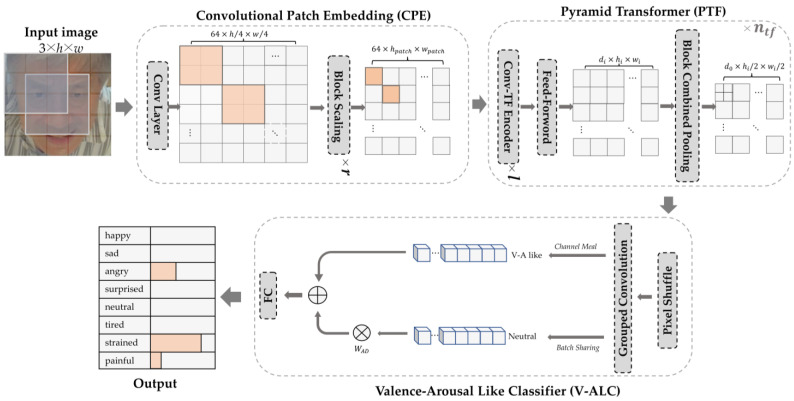
The overall architecture of the FER-PCVT, where r, l, and $n_{tf}$ are the numbers of repetitions of the Block Scaling, the Conv-TF Encoder, and the Pyramid Transformer, respectively. HA: happy; SA: sad; AN: angry; SU: surprised; NE: neutral; TI: tired; ST: strained; PA: painful.

**Figure 4 brainsci-12-01626-f004:**
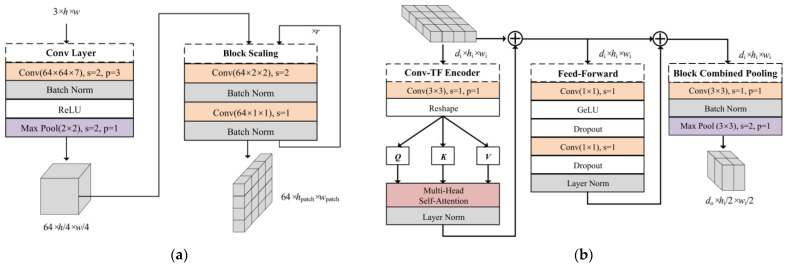
The pipeline of the PCVT architecture. (**a**) Details of the Convolutional Patch Embedding (CPE). (**b**) Details of the Pyramid Transformer (PTF).

**Figure 5 brainsci-12-01626-f005:**
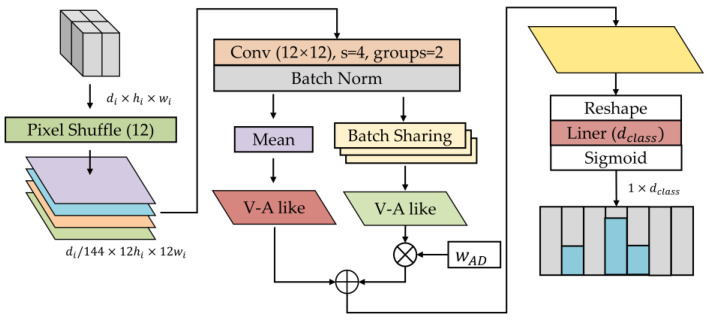
Details of the Valence-Arousal-Like Classifier (V-ALC).

**Figure 6 brainsci-12-01626-f006:**
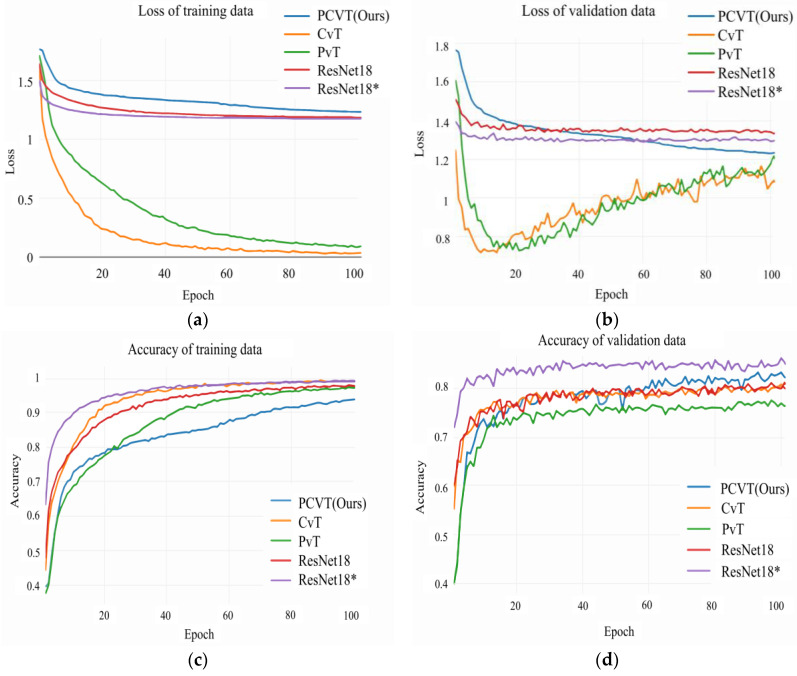
Iterative curves of CvT, PvT, ResNet18, ResNet18*, and PCVT on the RAF-DB dataset. (**a**) Loss plots of these five networks on the training data of RAF-DB. (**b**) Loss plots of these five networks on the validation data of RAF-DB. (**c**) Accuracy plots of these five networks on the training data. (**d**) Accuracy plots of these five networks on the validation data. * It represents a pretrained model.

**Figure 7 brainsci-12-01626-f007:**
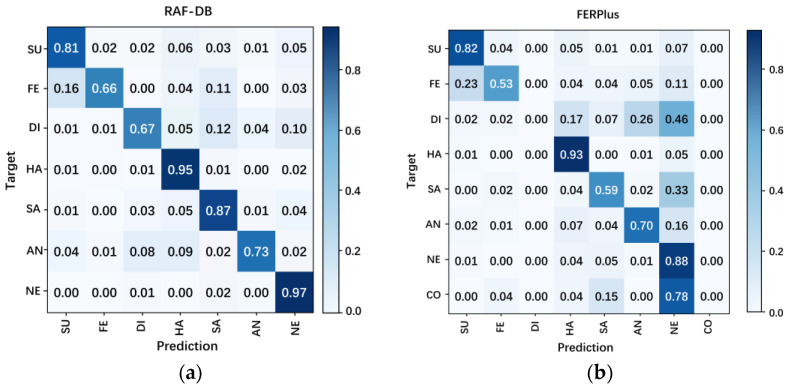
Confusion matrices of FER-PCVT on the RAF-DB dataset (**a**) and the FER+ dataset (**b**). SU: surprise; FE: fear; DI: disgust; HA: happy; SA: sad; AN: angry; NE: neutral; CO: contempt.

**Figure 8 brainsci-12-01626-f008:**
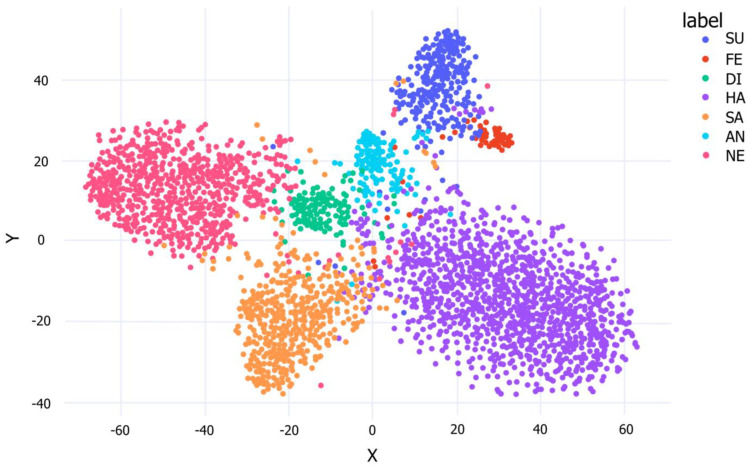
The t−SNE of FER-PCVT on the RAF-DB dataset. SU: surprise; FE: fear; DI: disgust; HA: happy; SA: sad; AN: angry; NE: neutral; CO: contempt.

**Figure 9 brainsci-12-01626-f009:**
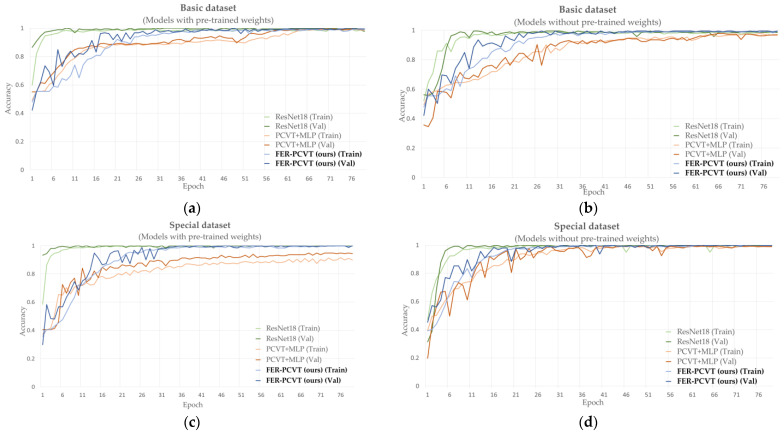
Training and validation accuracy curves of ResNet18, PCVT+MLP, and FER-PCVT on the facial expression dataset of stroke patients. (**a**) Accuracy curves for models with pretrained weights on the basic dataset; (**b**) accuracy curves for models without pretrained weights on the basic dataset; (**c**) accuracy curves for models with pretrained weights on the special dataset; (**d**) accuracy curves for models without pretrained weights on the special dataset. Among them, “Train” means training, and “Val” means validation, and the bold font indicates the algorithm proposed in this study.

**Figure 10 brainsci-12-01626-f010:**
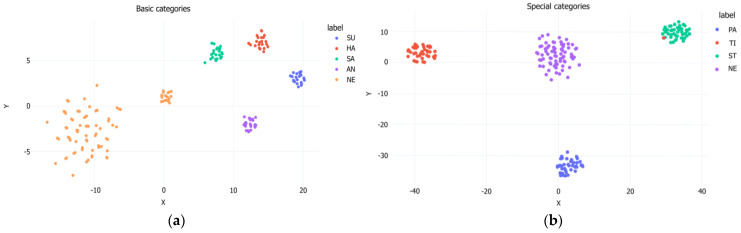
The t−SNE of FER-PCVT on the private dataset. (**a**) Visualization of the clustering performance of FER-PCVT for five basic expressions; (**b**) Visualization of the clustering performance of FER-PCVT for four special expressions. SU: surprised; HA: happy; SA: sad; AN: angry; NE: neutral; PA: painful; TI: tired; ST: strained.

**Figure 11 brainsci-12-01626-f011:**
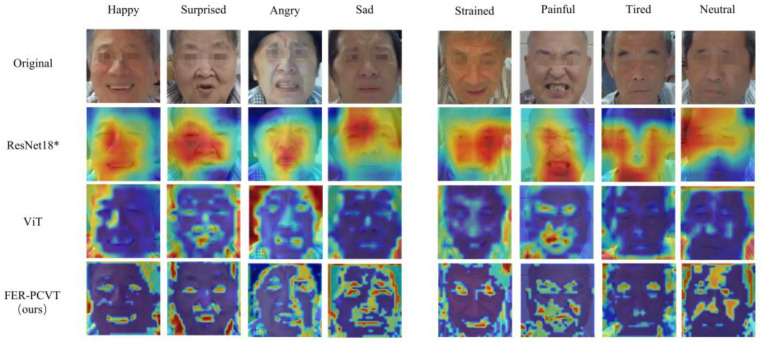
Visualization of ResNet18*, ViT, and FER-PCVT. * It indicates that the model is a pretrained model.

**Table 1 brainsci-12-01626-t001:** Properties of three datasets with data.

Class	FER+		RAF-DB		Private Dataset	
	Simple Size	Proportion (%)	Simple Size	Proportion (%)	Simple size	Proportion (%)
happy	5165	24.41	5957	38.84	141	10.83
surprised	3963	18.73	1619	10.55	62	4.76
sad	3765	17.79	2460	16.04	78	5.99
angry	2594	12.26	867	5.65	44	3.38
neutral	4748	22.44	3204	20.89	509	39.09
fearful	633	2.99	355	2.31	-	-
disgusted	145	0.69	877	5.72	-	-
contempt	148	0.70	-	-	-	-
painful	-	-	-	-	85	6.53
strained	-	-	-	-	298	22.89
tired	-	-	-	-	85	6.53
Min/Max sample size	0.0281	-	0.0596	-	0.0864	-
SUM	21,161	100	15,339	100	1302	100

**Table 2 brainsci-12-01626-t002:** FACS definitions of eight expressions.

Emotions	Code *
painful	AU4 + (AU6/AU7) + (AU9/AU10) + AU43
strained	AU4 + AU6 + (AU23/AU24/AU28)
tired	AU43 + AU54
neutral	/
happy	AU6 + AU12
sad	AU1 + AU4 + AU15
surprised	AU1 + AU2 + AU5 + AU26
angry	AU4 + AU5 + AU7 + AU23

* AU1: Inner Brow Raiser; AU2: Outer Brow Raiser; AU4: Brow Lowerer; AU5: Upper Lid Raiser; AU6: Cheek Raiser; AU7: Lid Tightener; AU9: Nose Wrinkler; AU10: Upper Lip Raiser; AU12: Lip Corner Puller; AU15: Lip Corner Depressor; AU23: Lip Tightener; AU24: Lip Pressor; AU26: Jaw Drop; AU28: Lip Suck; AU43: Eyes Closed; AU54: Head down.

**Table 3 brainsci-12-01626-t003:** Training parameter settings.

Parameter	Setting
optimizer	AdamW [43] ^1^
loss function	the Cross Entropy Loss
batch size	120
epoch	200
learning rate	0.0003
exponential LR	0.99

^1^ The optimizer selected AdamW from the Adam series commonly used in training ViT [44].

**Table 4 brainsci-12-01626-t004:** Structural parameters for each module.

Module	Structural Params	Internal Params	Input Size	Output Size
CPE	r=1	patch size ^1^ = 8	3 × 128 × 128	64 × 16 × 16
PTF1 ^2^	l=2	$d_{0}$ = 192, heads ^3^ = 8	64 × 16 × 16	192 × 8 × 8
PTF2 ^2^	l=4	$d_{0}$ = 576, heads ^3^ = 4	192 × 8 × 8	576 × 4 × 4
V-ALC	-	$d_{class}$ = 7	576 × 4 × 4	1 × 7

^1^ The patch size is the size of each patch when the image is split into patches. ^2^ The PTF module is repeated twice in the model’s overall structure, i.e., $n_{tf}=2,$ so the PTF1 and PTF2 refer to the first and second times, respectively. ^3^ The heads are the setting of the Multi-Head Self-Attention in the PTF module.

**Table 5 brainsci-12-01626-t005:** Training results of five networks on the RAF-DB dataset.

Model	Params (M)	GFLOPs	Accuracy (%)
ResNet18	11.20	0.29	81.52
ResNet18 *	11.20	0.29	86.28
CvT	19.55	0.66	81.45
PvT	6.25	0.14	77.80
PCVT(Ours)	2.46	0.12	84.22

* It represents a pretrained model.

**Table 6 brainsci-12-01626-t006:** Performance comparison of FER-PCVT and recent FER models.

Model	Tags	Year	Accuracy	
			FER+	RAF-DB
SPWFA-SE [45]	CNN	2020	-	86.31%
RAN [29]	ResNet	2019	89.16%	86.90%
Ad-Corre [46]	CNN	2022		86.96%
DACL [28]	ResNet	2021	-	87.78%
VTFF [34]	ViT	2022	88.81%	88.14%
SCN [47]	CNN	2020	89.35%	88.14%
FER-VT [48]	ViT	2021	90.04%	88.26%
PSR [49]	VGG-16	2020	-	88.98%
RUL [50]	ResNet	2021	-	88.98%
LResNet50E-IR [51]	ResNet	2020	89.257%	89.075%
FER-PCVT(Ours)	ViT	2022	88.21%	89.44%

**Table 7 brainsci-12-01626-t007:** Precision, Specificity, Sensitivity, F1-score, and G-mean of FER-PCVT on the Raf-DB and FER+ datasets.

Class	RAF-DB					FER+				
	Precision	Specificity	Sensitivity	F1-Score	G-Mean	Precision	Specificity	Sensitivity	F1-Score	G-Mean
surprised	0.877	0.987	0.81	0.842	0.894	0.912	0.982	0.82	0.864	0.898
fear	0.824	0.996	0.66	0.733	0.811	0.555	0.987	0.53	0.542	0.723
disgust	0.688	0.982	0.67	0.679	0.811	0	1	0	0	0
happy	0.941	0.962	0.95	0.945	0.956	0.861	0.951	0.93	0.894	0.940
sad	0.864	0.974	0.87	0.867	0.921	0.835	0.975	0.59	0.691	0.758
angry	0.895	0.995	0.73	0.804	0.852	0.462	0.886	0.70	0.557	0.788
neutral	0.882	0.966	0.97	0.924	0.968	0.632	0.851	0.88	0.736	0.865
contempt	-	-	-	-		0	1	0	0	0

**Table 8 brainsci-12-01626-t008:** Accuracy comparison on the facial expression dataset of stroke patients.

Model	Params (M)	Pre-Training	Accuracy (%)		
			Basic Categories ^1^	Special Categories ^2^	AVG
ResNet18	11.19	✘	98.72	99.66	99.19
		✔	99.58	99.72	99.65
RCVT+MLP	4.06	✘	88.46	97.22	92.84
		✔	98.64	99.21	98.93
PCVT+V-ALC (Ours)	4.10	✘	99.15	99.42	99.29
		✔	99.89	99.72	99.81

^1^ The basic categories include surprised, happy, sad, and angry expressions. ^2^ The special categories include tired, neutral, strained, and painful expressions.

## Data Availability

Two publicly available datasets FER+ and RAF-DB were analyzed in this study. The FER+ dataset can be found here: Challenges in Representation Learning: Facial Expression Recognition Challenge|Kaggle. The RAF-DB dataset can be found here: Real-world Affective Faces (RAF) Database (whdeng.cn). The private datasets in this study are available from the corresponding authors upon reasonable request.

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
