# Peer review of "FER-PCVT: Facial Expression Recognition with Patch-Convolutional Vision Transformer for Stroke Patients"

_brainsci, 2022, doi:10.3390/brainsci12121626_

Round 1

Reviewer 1 Report

The study presented by the authors seems very interesting to me, however there are some recommendations that should be taken into account before publication.

First of all, the tables and figures are not inserted correctly and make reading difficult, they need to be edited correctly.

The study presented by the authors seems very interesting to me, however there are some recommendations that should be taken into account before publication.

First of all, the tables are not inserted correctly and make reading difficult, they need to be edited correctly.

In the introduction of lines 59 to 76 there is no bibliographic reference associated with said information.

It would be interesting if you could include a paragraph in your discussion on how this type of technique would improve stroke detection, what clinical repercussions it could have.

It would be advisable to summarize the methods and results, sometimes they are too long and make the reader lose interest in what is shown.

Reviewer 2 Report

This study aimed to propose a DL approach for facial expression recognition for stroke patients. I have the following major suggestions.

What is the novelty of this study although several DL approach for facial expression recognition for stroke patients have been studied earlier?

Please write down the contribution of the study at the end part of the Introduction section in bulleted form.

Authors should introduce the ML/DL numerous applications in broad ranges, such as mental workload, disease prediction, stress, emotion. Stroke prediction studies should be explored in the Introduction section. Brainwaves are investigated for stroke prediction in the article, healthsos: real-time health monitoring system for stroke prognostics and in article, quantitative evaluation of task-induced neurological outcome after stroke. Biosignals are investigated for stroke prediction in article, big-ecg: cardiographic predictive cyber-physical system for stroke management; and in article, prediction of myoelectric biomarkers in post-stroke gait.

Authors should include conceptual figures of their DL proposed frameworks with more details and model parametrization. Figure quality needs to improve.

Authors should present the training and validation accuracy graphs of the proposed model with changes in the number of images.

Authors should report more performance measures of prediction models, such as sensitivity, specificity, precision, and negative predictive value from the confusion matrix.

Authors should discuss the strength and weaknesses of the proposed method/findings with other recent studies in the discussion section.

From the writing point of view, the manuscript must be checked for typos and the grammatical issues should be improved.

Round 2

Reviewer 1 Report

Figure 2 still does not have anything of quality to read it, the authors have not modified this point.

The images still do not have quality.

The article is too long, after a while it becomes boring and heavy to read.

It would be convenient to make a synthesis of what the authors exposed.

Reviewer 2 Report

Thanks for addressing the comments.

Figures quality need to be improved. 
